# Impact of TSPO Receptor Polymorphism on [^18^F]GE-180 Binding in Healthy Brain and Pseudo-Reference Regions of Neurooncological and Neurodegenerative Disorders

**DOI:** 10.3390/life11060484

**Published:** 2021-05-26

**Authors:** Franziska J. Vettermann, Stefanie Harris, Julia Schmitt, Marcus Unterrainer, Simon Lindner, Boris-Stephan Rauchmann, Carla Palleis, Endy Weidinger, Leonie Beyer, Florian Eckenweber, Sebastian Schuster, Gloria Biechele, Christian Ferschmann, Vladimir M. Milenkovic, Christian H. Wetzel, Rainer Rupprecht, Daniel Janowitz, Katharina Buerger, Robert Perneczky, Günter U. Höglinger, Johannes Levin, Christian Haass, Joerg C. Tonn, Maximilian Niyazi, Peter Bartenstein, Nathalie L. Albert, Matthias Brendel

**Affiliations:** 1Department of Nuclear Medicine, University Hospital of Munich, LMU Munich, 81377 Munich, Germany; Franziska.Vettermann@med.uni-muenchen.de (F.J.V.); Stefanie.Harris@med.uni-muenchen.de (S.H.); Julia.Schmitt@med.uni-muenchen.de (J.S.); Simon.Lindner@med.uni-muenchen.de (S.L.); Leonie.Beyer@med.uni-muenchen.de (L.B.); Florian.Eckenweber@med.uni-muenchen.de (F.E.); Sebastian.Schuster@med.uni-muenchen.de (S.S.); Gloria.Biechele@med.uni-muenchen.de (G.B.); Ferschi@gmx.net (C.F.); Peter.Bartenstein@med.uni-muenchen.de (P.B.); Nathalie.Albert@med.uni-muenche.de (N.L.A.); 2Department of Radiology, University Hospital of Munich, LMU Munich, 81377 Munich, Germany; Marcus.Unterrainer@med.uni-muenchen.de (M.U.); Boris.Rauchmann@med.uni-muenchen.de (B.-S.R.); 3Department of Psychiatry and Psychotherapy, University Hospital of Munich, LMU Munich, 81377 Munich, Germany; Robert.Perneczky@med.uni-muenchen.de; 4Department of Neurology, University Hospital of Munich, LMU Munich, 81377 Munich, Germany; Carla.Palleis@med.uni-muenchen.de (C.P.); Endy.Weidinger@med.uni-muenchen.de (E.W.); Johannes.Levin@med.uni-muenchen.de (J.L.); 5Department of Psychiatry and Psychotherapy, University of Regensburg, 93053 Regensburg, Germany; Vladimir.Milenkovic@ukr.de (V.M.M.); Christian.Wetzel@klinik.uni-regensburg.de (C.H.W.); Rainer.Rupprecht@medbo.de (R.R.); 6Institute for Stroke and Dementia Research, University Hospital of Munich, LMU Munich, 81377 Munich, Germany; Daniel.Janowitz@med.uni-muenchen.de (D.J.); Katharina.Buerger@med.uni-muenchen.de (K.B.); 7German Center for Neurodegenerative Diseases (DZNE), 81377 Munich, Germany; Hoeglinger.Guenter@mh-hannover.de (G.U.H.); Christian.Haass@mail03.med.uni-muenchen.de (C.H.); 8Ageing Epidemiology (AGE) Research Unit, School of Public Health, Imperial College, London SW7 2AZ, UK; 9Munich Cluster for Systems Neurology (SyNergy), 81377 Munich, Germany; 10Department of Neurology, Hannover Medical School, 30625 Hannover, Germany; 11Chair of Metabolic Biochemistry, Biomedical Center (BMC), Faculty of Medicine, LMU Munich, 82152 Planegg, Germany; 12Department of Neurosurgery, University Hospital of Munich, 81377 Munich, Germany; Joerg.Christian.Tonn@med.uni-muenchen.de; 13Department of Radiation Oncology, University Hospital of Munich, LMU Munich, 81377 Munich, Germany; Maximilian.Niyazi@med.uni-muenchen.de; 14German Cancer Consortium (DKTK), Partner Site Munich, 81377 Munich, Germany

**Keywords:** microglia, neurodegeneration, Alzheimer’s disease, neuro-oncology, 4R-tauopathy, TSPO-PET

## Abstract

TSPO-PET tracers are sensitive to a single-nucleotide polymorphism (rs6971-SNP), resulting in low-, medium- and high-affinity binders (LABs, MABs and HABS), but the clinical relevance of [^18^F]GE-180 is still unclear. We evaluated the impact of rs6971-SNP on in vivo [^18^F]GE-180 binding in a healthy brain and in pseudo-reference tissue in neuro-oncological and neurodegenerative diseases. Standardized uptake values (SUVs) of [^18^F]GE-180-PET were assessed using a manually drawn region of interest in the frontoparietal and cerebellar hemispheres. The SUVs were compared between the LABs, MABs and HABs in control, glioma, four-repeat tauopathy (4RT) and Alzheimer’s disease (AD) subjects. Second, the SUVs were compared between the patients and controls within their rs6971-subgroups. After excluding patients with prior therapy, 24 LABs (7 control, 5 glioma, 6 4RT and 6 AD) were analyzed. Age- and sex-matched MABs (n = 38) and HABs (n = 50) were selected. The LABs had lower frontoparietal and cerebellar SUVs when compared with the MABs and HABs, but no significant difference was observed between the MABs and HABs. Within each rs6971 group, no SUV difference between the patients and controls was detected in the pseudo-reference tissues. The rs6971-SNP affects [^18^F]GE-180 quantification, revealing lower binding in the LABs when compared to the MABs and HABs. The frontoparietal and cerebellar ROIs were successfully validated as pseudo-reference regions.

## 1. Introduction

The translocator protein 18 kDa (TSPO), previously known as the peripheral benzodiazepine receptor, is a mitochondrial transporter involved in various intracellular processes. Its expression in the central nervous system (CNS) under physiological conditions is relatively low, but expression is upregulated in activated microglia, macrophages and cancer cells [1]. TSPO has become more important as a positron emission tomography (PET) imaging target for several diseases, including CNS autoimmune diseases, neurodegeneration and glioma [2].

The first-generation TSPO-PET tracer [^11^C]PK11195 has been used for over 25 years, though its application is limited due to poor pharmacokinetics and carbon-11 radiolabeling [3,4,5]. Several second-generation TSPO ligands with an improved signal-to-noise ratio, including [^11^C]PBR28, have been investigated and revealed to have substantial heterogeneity in their binding potentials due to inter-subject variability in the affinity for TSPO [6,7]. Here, the binding properties of second-generation TSPO ligands were found to depend on a genetic polymorphism in the TSPO gene. A single-nucleotide polymorphism (rs6971) that replaces alanine with threonine (Ala147Thr) results in three patterns of binding affinity: high-affinity binders (HABs), medium-affinity binders (MABs) and low-affinity binders (LABs), depending on the homozygosity or heterozygosity of the allele [8]. A loss of binding to TSPO in approximately 10% of the LABs and underestimation of TSPO expression in the LABs and MABs were reported [9]. Similar effects of the rs6971 polymorphism were also demonstrated in initial studies using [^11^C]PBR28 [10,11], and this phenomenon was subsequently noticed for several second-generation radiotracers [6,11,12]. Thus, the PET signals of patients with the MAB or LAB status significantly underestimate TSPO expression, requiring the TSPO binding status to be determined. The development of next-generation TSPO tracers was a consequence of the sensitivity of the second-generation tracers to this polymorphism in the TSPO gene. Recently, the next-generation TSPO-PET tracer [^18^F]GE-180 has received attention due to its fluorine labeling, which makes the tracer available at centers without an on-site cyclotron. Furthermore, recent studies reported a high lesion-to-background ratio and a higher proportion of specific binding (45%) when compared to [^11^C]PBR28 (33%), as shown by in vivo blocking [13,14,15,16].

Our group has studied [^18^F]GE-180 in neuro-oncological, neurodegenerative and neuroimmune diseases. Glioma imaging with [^18^F]GE-180 has been shown to be valuable in noninvasive grading, with excellent sensitivity for the detection of high-grade gliomas [17,18,19]. [^18^F]GE-180 imaging in patients with a clinical diagnosis of 4R-tauopathy (4RT) closely reflected the expected topology of microglial activation and indicated early detection in the disease’s course [20]. These findings were underpinned by strong immunohistochemical correlations of CD68 staining and TSPO-PET signals in a Trem2-deficient amyloid mouse model and in tau transgenic P301S mice [21,22]. However, the clinical relevance of the rs6971 polymorphism to different levels of binding affinity has not yet been systematically determined for [^18^F]GE-180.

In this study, we aimed to evaluate the impact of rs6971 on the in vivo [^18^F]GE-180 signal in a healthy brain and in potential pseudo-reference tissue for studies on neuro-oncological and neurodegenerative diseases. We hypothesized that [^18^F]GE-180 would be sensitive to the rs6971 polymorphism due to the tracer’s specificity to TSPO.

## 2. Results

### 2.1. Sample Composition and Demographics

From a total of 380 participants, 113 participants were included in the analysis. Genotyping revealed 12% LABs (45/380), and after applying exclusion criteria, 24 LABs were included: 7 control subjects, 5 patients with glioma, 6 patients with a clinical diagnosis of a 4R-tauopathy and 6 patients with a clinical diagnosis of AD, with a mean age of 67.9 years (95% CI: 64.6–71.8) and a female-to-male ratio of 12:12. The MAB and HAB patients were selected by a matching algorithm including the minimal n, age and sex (Figure 1).

There were also 38 MABs included: 14 patients with glioma, 8 patients with a clinical diagnosis of a 4R-tauopathy, 6 patients with a clinical diagnosis of AD and 10 control patients, with a mean age of 70.1 years (95% CI: 67.2–72.9) and a female-to-male ratio of 23:15. Finally, 52 HABs were included: 11 patients with glioma, 21 with a clinical diagnosis of a 4R-tauopathy, 15 with a clinical diagnosis of AD and 5 control subjects, with a mean age of 70.4 (95% CI: 67.9–72.8) and a female-to-male ratio of 23:29 (Table 1). The diagnosis-specific subgroups did not differ in binding status, age or sex. The glioma group consisted of 27 WHO grade IV gliomas with an IDH-wildtype status and one WHO grade II glioma with IDH mutation. As previously described, to assess the maximal tumor-to-background ratio (TBR_max_), the maximum standardized uptake value (SUV_max_) of the tumor was divided by the background [18]. The SUV_max_ was 2.8 (95% CI: 2.5–3.1), and the maximum tumor-to-background ratio (TBR_max_) was 6.6 (95% CI: 5.8–7.5). The analyzed patients with 4R-tauopathies had a PSPRS score of 29.8 (95% CI: 24.8–34.8), a MoCA score of 22.4 (95% CI: 20.6–24.2) and a SEADL score of 60.6 (95% CI: 54.2–67.1). The analyzed patients on the AD continuum had an MMSE score of 24.1 (95% CI: 21.9–26.2), a global CDR score of 0.6 (95% CI: 0.39–0.76) and a CDR-SOB score of 3.2 (95% CI: 2.1–4.3). The controls did not show any signs of cognitive decline (MoCA 29.1 (95% CI: 28.4–29.7)) or motor dysfunction.

### 2.2. [^18^F]GE-180 Binding in a Comparison of rs6971 Polymorphism Subgroups

Concordance between repeated quantifications by defined manual regions was excellent for the frontoparietal (r = 0.987) and cerebellar (r = 0.984) VOI. Overall, the LABs showed significantly lower SUVs in the frontoparietal and cerebellar VOIs compared with the MABs and HABs (Table 2). There was no significant difference observed between the MABs and HABs (Table 2). Specific findings in the control and disease groups are reported below.

#### 2.2.1. Controls

The frontoparietal SUV of the LAB controls was significantly lower (0.345 ± 0.025) compared with the MAB controls (0.436 ± 0.021, *p* = 0.013) and HAB controls (0.471 ± 0.031, *p* = 0.006). A comparable effect was detected in the cerebellum, with a significantly lower SUV for the LAB controls (0.367 ± 0.027) compared with the MAB controls (0.455 ± 0.023, *p* = 0.023) and HAB controls (0.514 ± 0.034, *p* = 0.004). There was no discernible difference in the SUVs between the MAB and HAB controls in both regions (frontoparietal, *p* = 0.375; cerebellum, *p* = 0.181; Figure 2).

#### 2.2.2. Disease Groups

##### Glioma

The frontoparietal and cerebellar SUVs in the glioma cohort differed significantly between the three TSPO binding polymorphism groups. The frontoparietal VOI of the LABs revealed significantly lower SUVs (0.381 ± 0.021) compared with the MABs (0.424 ± 0.012, *p* = 0.086) and HABs (0.436 ± 0.014, *p* = 0.047).

The cerebellar VOI in the glioma group showed similar results, with lower SUVs in the LABs (0.372 ± 0.028) compared with the MABs (0.463 ± 0.016, *p* = 0.009) and a trend toward lower SUVs in the HABs (0.440 ± 0.018, *p* = 0.059). Again, no difference in SUVs between the MAB and HAB glioma patients in both regions was detectable (frontoparietal, *p* = 0.551; cerebellum, *p* = 0.338).

##### 4RT

In patients with 4R-tauopathy, we observed a significantly lower frontoparietal SUV in the LABs (0.355 ± 0.028) compared with the MABs (0.455 ± 0.024, *p* = 0.010) and HABs (0.462 ± 0.015, *p* = 0.002). In the cerebellum, the SUV was significantly lower in the LABs (0.354 ± 0.032) compared with the MABs (0.475 ± 0.028, *p* = 0.007) and HABs (0.478 ± 0.017, *p* = 0.002). There was no difference between the SUVs in the MAB and HAB 4R-tauopathy patients (frontoparietal, *p* = 0.826; cerebellar, *p* = 0.930).

##### AD

The results for patients with AD trended in the same direction as the other disease groups but did not reach statistical significance. The frontoparietal SUV of the LABs was 0.360 ± 0.039, compared with 0.420 ± 0.039 (*p* = 0.290) for the MABs and 0.435 ± 0.025 (*p* = 0.118) for the HABs. In the cerebellum, an insignificantly lower SUV was observed in the LABs (0.397 ± 0.038), compared with the MABs (0.452 ± 0.038, *p* = 0.321) and HABs (0.460 ± 0.024, *p* = 0.175). There was no discernible difference in SUVs between the MAB and HAB AD patients for both regions (frontoparietal, *p* = 0.740; cerebellum, *p* = 0.848).

### 2.3. [^18^F]GE-180 Binding in Frontoparietal and Cerebellar Pseudo-Reference Tissues in the rs6971 Polymorphism Subgroups

Visually, within each rs6971 polymorphism group, tracer binding of the frontoparietal and cerebellar VOIs was comparable between the patients and healthy controls (Figure 3).

#### 2.3.1. LABs

Within the LABs, the frontoparietal SUV as a potential pseudo-reference tissue for glioma TSPO-PET imaging was similar between patients with glioma and the controls (*p* = 0.599). Similar results were obtained for cerebellar SUVs, which can be applied as a pseudo-reference tissue for TSPO-PET imaging of patients with glioma and neurodegenerative diseases. No significant difference was detected between all patients and control subjects (glioma vs. 4R-tauopathy, *p* = 0.817; glioma vs. AD, *p* = 0.174; glioma vs. control, *p* = 0.837; 4R-tauopathy vs. AD, *p* = 0.093; 4R-tauopathy vs. control, *p* = 0.626; AD vs. control, *p* = 0.184).

#### 2.3.2. MABs and HABs

In the rs6971 polymorphism subgroups of MABs and HABs, no significant differences in the frontoparietal SUVs were detected between patients with glioma and the controls (*p* = 0.405, *p* = 0.908). The cerebellar SUVs revealed similar results, with no significant differences between all patients and control subjects regarding MABs and HABs (glioma vs. 4R-tauopathy, *p* = 0.791, *p* = 0.220; glioma vs. AD, *p* = 0.567, *p* = 0.244; glioma vs. control, *p* = 0.977, *p* = 0.355; 4R-tauopathy vs. AD, *p* = 0.470, *p* = 0.987; 4R-tauopathy vs. control, *p* = 0.784, *p* = 0.936; AD vs. control, *p* = 0.608, *p* = 0.946).

### 2.4. Overall Impact of the rs6971 Polymorphism on [^18^F]GE-180 Binding

Finally, we aimed to determine the quantitative impact of the rs6971 polymorphism on [^18^F]GE-180 SUV by a combined analysis of all the study groups. Limited pathology in the cerebellum of all included subjects and limited pathology in the frontoparietal region of the patients with glioma and the controls allowed us to estimate the impact of rs6971 polymorphism in a robust fashion. The averaged difference between the SUV of the LABs and the MABs was 19.0 ± 5.9%, and between the LABs and HABs, it was 20.9 ± 5.3% (Figure 4). The impact was robust when considering the quantitative coefficient of variance of all four study groups.

## 3. Discussion

In this study, we investigated the tracer binding of [^18^F]GE-180 in the three rs6971 polymorphism subgroups by analyzing healthy controls, patients with glioma and patients with neurodegenerative diseases. The aim was to determine if [^18^F]GE-180 was sensitive to the rs6971 polymorphism, as is known from other TSPO radioligands, and to evaluate potential pseudo-reference tissues for simplified clinical application of this tracer.

The main strength of our study is the meaningful sample of 24 LABs, after excluding relevant confounding factors such as radiotherapy, chemotherapy or steroid use. Overall, the LABs showed significantly lower [^18^F]GE-180 SUVs compared with the MABs and HABs, and the impact of the LAB status was estimated to be a reduction in the SUV of 19% and 21%, respectively. In a study using postmortem tissue for in vitro tracer binding, the TSPO-PET radioligands PBR28 and PBR06 showed a more pronounced difference in binding affinity, with 50- and 17-fold higher affinities of HABs compared with the LABs, respectively. PBR111 and DPA713 indicated a 4-fold higher affinity of the HABs compared with the LABs, and PK11195 showed a negligible difference in binding affinity (0.8-fold) of the HABs compared with the LABs [6,23]. The purpose of our study was to evaluate the impact of the rs6971 polymorphism on [^18^F]GE-180 application in vivo. We note that head-to-head studies would be required for a direct comparison of the rs6971 polymorphism’s impact on image quantification between different tracers. However, in line with many of the aforementioned studies, we found a sensitivity of this tracer for the rs6971 polymorphism. In conclusion, LABs need to be considered carefully when performing TSPO-PET imaging with [^18^F]GE-180 in neuro-oncology and neurology, since the rs6971 polymorphism significantly impacts quantification. From the current perspective, LABs should be excluded, and it should be feasible to include MABs and HABs when considering the rs6971 polymorphism as a covariate, given the minor differences in binding between MABs and HABs.

The validity of [^18^F]GE-180 as an in vivo read out of microglial activation has been discussed extensively, with a focus on the question of whether [^18^F]GE-180 uptake is mainly driven by blood–brain barrier disruption [24,25,26,27]. In this regard, [^18^F]GE-180 revealed a high signal-to-noise ratio in preclinical studies but low brain penetration in the human healthy controls [15,28]. A lack of rs6971 polymorphism differences in diseases [17,29] has been one major argument questioning the specificity of [^18^F]GE-180 binding to the microglial TSPO receptor. Our study demonstrates for the first time a significant impact of rs6971 polymorphism on [^18^F]GE-180 uptake in vivo, thus refuting the earlier criticism [26]. In line with our data, Sridharan et al. performed a blocking study to quantify the specific binding of [^18^F]GE-180 to TSPO and measured a 45% specific signal, concluding that despite low brain penetration, [^18^F]GE-180 exhibits a specific signal in the brain [16]. Furthermore, our translational study in P301S mice and patients with 4R-tauopathy indicated microglia-specific uptake of [^18^F]GE-180 in a depletion experiment and no dependency of the tracer uptake by markers of blood–brain barrier integrity. Microleakage was claimed to be a potential source of [^18^F]GE-180 signal elevation in regions without MRI contrast enhancement [18,19,26,30]. Our current data did not show an elevated [^18^F]GE-180 signal in the pseudo-reference regions of patients with glioma or neurodegenerative diseases when compared with the healthy controls, regardless of the rs6971 polymorphism status. Thus, a general disease-related microleakage as the main driver of the [^18^F]GE-180 signal seems unlikely. In summary, our data enhance the evidence for a specific [^18^F]GE-180 signal in vivo.

The second analysis of the current study questioned the use of potential pseudo-reference regions for [^18^F]GE-180 PET imaging. Molecular imaging in neuro-oncology is most commonly performed with amino acid tracers, such as [^18^F]FET. Our group evaluated the frontoparietal hemisphere as favorable pseudo-reference tissue for the quantification of [^18^F]FET tumor uptake in clinical routines [28]. However, a concomitant neuroinflammatory response in brain regions without tumor infiltration could potentially affect TSPO-PET binding in the contralateral pseudo-reference tissue. Here, [^18^F]GE-180 SUVs in patients with glioma were not different compared to the healthy controls within each polymorphism subgroup. Thus, the [^18^F]GE-180 SUVs did not indicate that the presence of glioma in the contralateral hemisphere had any impact. Studies with other TSPO radioligands reported that TSPO expression in gliomas is predominantly related to neoplastic cells and a lack of TSPO expression in surrounding reactive astrocytes, and we note that further research using step-wise stereotactic biopsies in spatial correlation with PET is warranted to further elucidate the issue of general neuroinflammation in the presence of glioma [31,32,33]. Taken together, our results support the use of the frontoparietal region as a pseudo-reference region for neurooncological imaging with [^18^F]GE-180 analogous to [^18^F]FET, at least for primary diagnosis prior to any therapeutic intervention.

The cerebellum is frequently used as pseudo-reference tissue in PET imaging of neurodegenerative diseases due to the low disease burden and limited β-amyloid and tau pathology in postmortem samples of AD and 4R tauopathies, with the exception of the dentate nucleus [34,35,36]. In addition, our current study did not indicate significant differences in [^18^F]GE-180 SUVs in the cerebella of patients with neurodegenerative diseases when compared to the controls within each polymorphism subgroup. This qualifies the cerebellum as a suitable pseudo-reference region for TSPO-PET imaging of neurodegenerative diseases using [^18^F]GE-180. Along the same lines, others have also suggested the cerebellum as a suitable pseudo-reference region for [^11^C]PBR28 imaging of AD [37]. Taken together, our [^18^F]GE-180 analysis of dedicated rs6971 polymorphism subgroups revealed that potential pseudo-reference tissues for neurooncological and neurodegenerative diseases did not show altered binding in diseases when compared to the controls, thus making them suitable for relative quantification. Of note, we also did not find significant increases in [^18^F]GE-180 binding in the frontoparietal VOIs of patients with AD or 4R-tauopathy when compared to the healthy controls, although microglial activation in these cortical areas is known in AD and 4RT. However, our previous work in 4R-tauopathy indicated that signal elevation of cortical TSPO-PET is phenotype-dependent and regionally heterogeneous in individual patients, with maximum VOI differences of ~15% at the group level [38]. Thus, it was not surprising that the less robust measure of SUV (in terms of variance) did not deliver significant group differences between 4RT or AD and the controls in a manually drawn frontal cortical VOI. We must note that quantitative measures in studies of neurodegenerative disorders commonly use a more robust SUV ratio instead of SUVs. However, we used an SUV in line with our previous work in the neurooncological field with the established pseudo-reference tissue of glioma imaging (frontoparietal crescent-shaped gray matter and white matter VOI), since it was not the purpose of our study to quantify TSPO expression in disease-specific target regions. It remains to be tested if pseudo-reference region normalization facilitates the concomitant use of LAB data together with MAB and HAB data in diseases. This will depend on the presence or absence of disease-specific binding in the target regions of LABs, and the investigation will require similar patient cohorts for the rs6791 polymorphism subgroup.

Limitations of the study need to be considered. Our results were based on static [^18^F]GE-180 scans acquired from 60 to 80 min p.i. without arterial blood sampling. Thus, we were not able to consider the impact of kinetic modeling, tracer plasma availability or tracer metabolism in different rs6971 polymorphism subgroups. However, given the low incidence of LABs, we note that large numbers of participants with dynamic scanning or arterial sampling would be required to address such questions. In this regard, stable time–activity curves after 30 min p.i. have been shown in healthy tissues in former studies [19,30]. Another limitation is the small sample size of several of our subgroups, though this is also related to the generally low incidence of LABs. Thus, the statistical power was not sufficient to include several covariates that have been shown to be associated with TSPO-PET binding, such as obesity [39].

The rs6971 polymorphism has an impact on [^18^F]GE-180 quantification in vivo, leading to 19% and 21% reductions, respectively, of the SUVs in LABs compared to MABs and HABs. The frontoparietal and cerebellar pseudo-reference regions can be used for patients with glioma and neurodegenerative diseases.

## 4. Methods

### 4.1. Study Design, Study Population and Clinical Assessments

The study and data analyses (ethics applications: 17-569, 17-755, 17-656 and 19-022) were approved by the local ethics committee (LMU, Munich, Germany). Genotyping was performed for all subjects who received a TSPO-PET scan at the Department of Nuclear Medicine of the University Hospital of Ludwig Maximilian University (LMU) of Munich. Due to the disproportional distribution of the rs6971 polymorphism, the controls and patients with glioma, Alzheimer’s disease (AD) and 4RT were first screened for LABs. Age- and sex-matched MABs and HABs for the controls and all disease groups were included in order to secure a homogeneous study population using a step-wise, demographic-guided selection algorithm. All glioma patients with prior therapy (chemotherapy or radiotherapy) were excluded a priori. All subjects with immunomodulatory therapy (steroids) were excluded a priori. Patients with newly diagnosed and later confirmed glioma were included in the glioma cohort. Diagnosis of 4R-tauopathy was made according to the revised Armstrong criteria of probable CBS or the Movement Disorders Society criteria of possible or probable PSP or possible PSP with predominant CBS [40,41]. The AD continuum patients were required to meet the criteria for typical AD, with mild cognitive impairment or dementia according to the diagnostic criteria of the National Institute on Aging and the Alzheimer’s Association [41]. The exclusion criteria were severe neurological or psychiatric disorders other than AD continuum or 4R-tauopathy. The healthy controls had no evidence of cognitive impairment, based on a neuropsychological battery including the ADAS-Cog, a CDR score of 0, no family history of AD or neurological disease associated with dementia and no objective motor symptoms. For all available MAB and HAB cases, the algorithm excluded mismatched cases (in terms of LAB matching) until the age and sex were not different between rs6971 groups (*p* > 0.1). The algorithm was stopped for a group size of ≤5.

### 4.2. TSPO Genotyping

All individuals underwent genotyping for the genetic polymorphism of the TSPO gene and were classified as LAB, MAB or HAB. As was previously described [19], whole-blood samples were sent to the Department of Psychiatry of the University Hospital Regensburg for polymorphism genotyping. Genomic DNA was extracted from 4 mL of whole blood using a QIAamp DNA Blood Maxi Kit (Qiagen, Hilden, Germany) according to the manufacturer’s protocol. The DNA quality was assessed by optical absorbance and gel electrophoresis. Exon 4 of the TSPO gene and the exon/intron junctions were amplified by PCR and sequenced using the Sanger method with the following primers: ex4-F-AGTTGGGCAGTGGGACAG and ex4-R-GCAGATCCTGCAGAGACGA. Sequencing data were analyzed using SnapGene software (GSL Biotech; http://snapgene.com, accessed on 10 May 2021).

### 4.3. TSPO-PET Acquisition and Analysis

TSPO-PET scans were acquired with a Biograph 64 PET/CT scanner (Siemens, Erlangen, Germany) at the Department of Nuclear Medicine, LMU Munich. A low-dose computed tomography scan preceded the PET acquisition and served as attenuation correction. Automated production of [^18^F]GE-180 was performed as described previously [42]. After injection of 189 ± 12 MBq [^18^F]GE-180, all patients and controls received a static 60–80 min p.i. scan [20]. The respective summation images were used for image analysis [18,19]. The images were reconstructed using an OSEM3D algorithm (8 iterations, 4 subsets, 4 mm Gauss). For each scan, standard corrections for attenuation, scatter, decay and random counts were applied.

According to the previously evaluated and published method for assessing the background activity of [^18^F]FET in glioma [43], a merged VOI consisting of five manually drawn crescent-shaped ROIs in the frontoparietal lobe, including white and gray matter, was assessed on the contralateral side of the tumor.

For patients with neurodegenerative diseases, a manually drawn cerebellar VOI (HC, 4R-tauopathy and AD) was used in order to avoid β-amyloid or tau-positive supratentorial regions. Five manually drawn ROIs containing ~2 mL each were drawn in a crescent shape in the posterior lobe of the cerebellum, excluding the vermis, anterior lobe, peduncle and flocculus to ensure a sufficient distance to any vessels and to exclude regions involved in the disease. All five ROIs were merged into a single VOI for both regions.

For the purpose of comparison, both VOIs were drawn in all groups. In order to evaluate intra-reader variability, each patient was evaluated five times by a single operator. Concordance between repeated quantifications by manual region definition was calculated. Group comparisons of [^18^F]GE-180 SUVs between patients with glioma, 4R-tauopathy or AD and the controls, as well as within each TSPO polymorphism group, were performed by analysis of variance (ANOVA) with a significance level of *p* < 0.05 using age and sex as covariates.

## Figures and Tables

**Figure 1 life-11-00484-f001:**
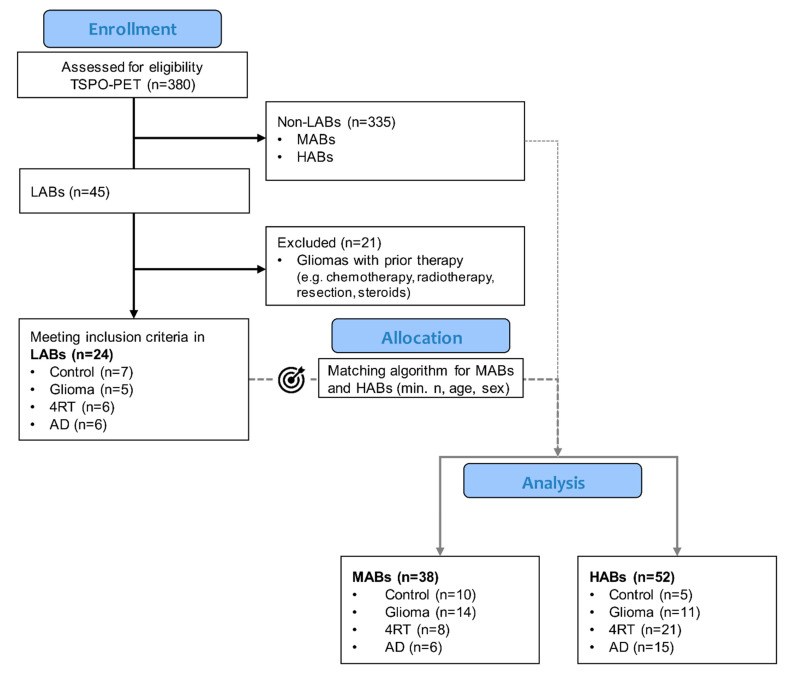
Flowchart of subject selection. LAB: low-affinity binder; MAB: medium-affinity binder; HAB: high-affinity binder; AD: Alzheimer’s disease; and 4RT: 4-repeat tauopathy.

**Figure 2 life-11-00484-f002:**
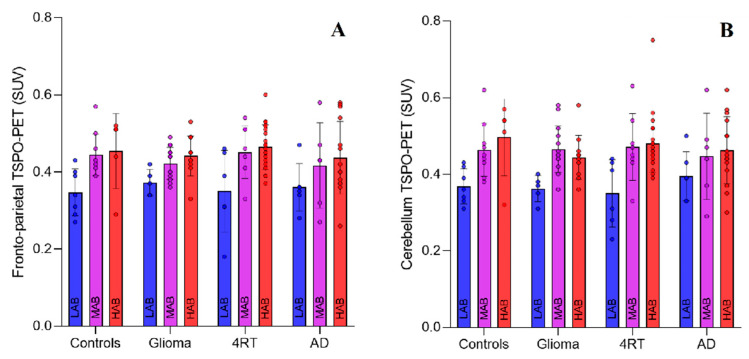
SUV_mean_ distribution among rs6971 polymorphism subgroups in the controls and neuro-oncological and neurodegenerative diseases for (**A**) frontoparietal and (**B**) cerebellar VOI. Error bars indicate standard deviations. LAB: low-affinity binder; MAB: medium-affinity binder; HAB: high-affinity binder; AD: Alzheimer’s disease; 4RT: 4-repeat tauopathy; and SUV: standardized uptake value.

**Figure 3 life-11-00484-f003:**
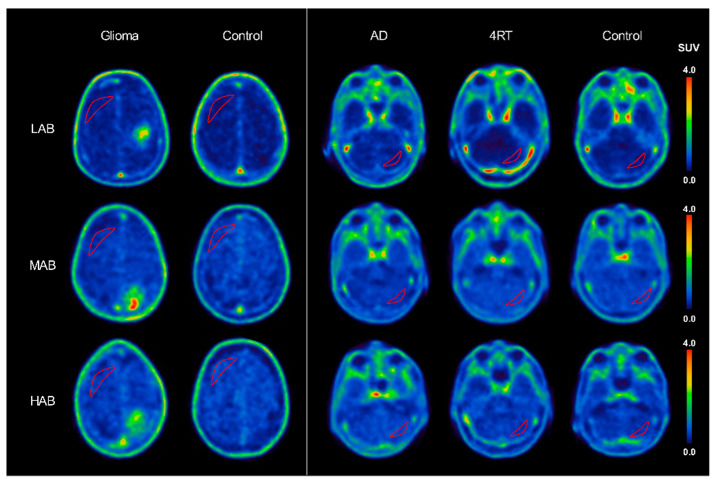
Varying [^18^F]GE-180 uptake in three polymorphism subgroups (LAB, MAB and HAB) but comparable intensity throughout the disease and control subjects within the rs6971 polymorphism subgroups. Crescent-shaped red lines represent drawn ROIs for pseudo-reference tissue assessment (as described in the Methods section). LAB: low-affinity binder; MAB: medium-affinity binder; HAB: high-affinity binder; AD: Alzheimer’s disease; 4RT: 4-repeat tauopathy; and SUV: standardized uptake value.

**Figure 4 life-11-00484-f004:**
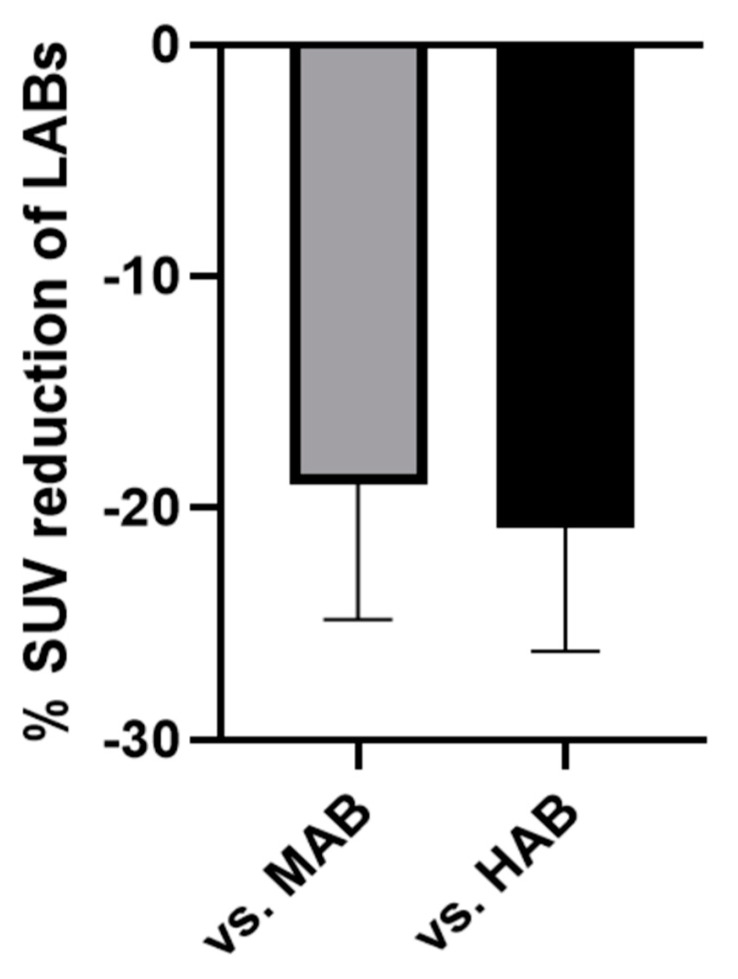
Percentage of [^18^F]GE-180 SUV reduction regarding genetically determined LAB status. Mean value ± standard deviation derived from six comparisons each (frontoparietal: HC and glioma; cerebellum: HC, glioma, 4RT and AD). LAB: low-affinity binder; MAB: medium-affinity binder; HAB: high-affinity binder; and SUV: standardized uptake value.

**Table 1 life-11-00484-t001:** Demographics at each group level.

		LAB	MAB	HAB
Number of subjects		24	38	52
Diagnosis	Control (n)	7	10	5
	Glioma (n)	5	14	11
	4R-tauopathy (n)	6	8	21
	AD (n)	6	6	15
Age (y, mean, 95%CI)		68.2	70.1	70.4
		(64.6–71.8)	(67.2–72.9)	(67.9–72.8)
Sex (♀/♂)		12♀/12♂	23♀/15♂	23♀/29♂
Control	Age (y, mean, 95% CI)	68.4	69.1	72.2
		(61.9–74.9)	(63.7–74.6)	(64.4–79.9)
	Sex (♀/♂)	4/3	7/3	1/4
Glioma	Age (y, mean, 95%CI)	60.9	70.0	71.9
		(51.7–70.0)	(64.6–75.5)	(65.7–78.1)
	Sex (♀/♂)	2/3	8/6	4/7
	SUV_max_	3.3	2.5	2.6
		(2.4–4.1)	(2.0–3.0)	(2.1–3.2)
	TBR_max_	8.7	6.2	6.2
		(6.9-10.5)	(5.0–7.3)	(4.9–7.5)
4RT	Age (y, mean, 95% CI)	69.9	68.1	68.9
		(62.4–77.3)	(61.7–74.6)	(64.9–72.8)
	Sex (♀/♂)	4/2	5/3	10/11
	PSPRS	23.0	33.8	28.9
		(3.0–42.9)	(23.8–35.1)	(22.7–35.1)
	MoCA	21.5	23.4	22.0
		(14.1–28.9)	(19.7–27.1)	(19.8–24.3)
	SEADL	75.0	56.3	61.0
		(49.3–100.7)	(43.4–69.1)	(53.0–68.9)
AD	Age (y, mean, 95% CI)	72.4	74.2	70.8
		(65.4–79.3)	(67.3–81.1)	(66.4–75.1)
	Sex (♀/♂)	2/4	3/3	8/7
	MMSE	23.4	23.3	24.6
		(18.3–28.5)	(18.7–27.9)	(21.7–27.5)
	CDR	0.50	0.75	0.53
		(0.06–0.94)	(0.35–1.15)	(0.28–0.79)
	CDR-SOB	3.8	4.6	2.5
		(1.4–6.2)	(2.4–6.8)	(1.1–3.9)

AD: Alzheimer’s disease; CI: confidence interval; MMSE: Mini-Mental State Examination; CDR: clinical dementia rating; SOB: sum of boxes; SEADL: Schwab and England activities of daily living; PSPRS: progressive supranuclear palsy rating scale; MoCA: Montreal cognitive assessment; 4RT: 4-repeat tauopathy; SUV_max_: maximum standardized uptake value; TBR_max_: maximum tumor-to-background ratio; LAB: low-affinity binder; MAB: medium-affinity binder; and HAB: high-affinity binder; ♀: female; ♂: male; y: years.

**Table 2 life-11-00484-t002:** Findings at a glance.

	Frontal-Parietal				Cerebellum			
Disease Group	LAB (SUV ± SD)	MAB (SUV ± SD)	HAB (SUV ± SD)	*p*-Value	LAB (SUV ± SD)	MAB (SUV ± SD)	HAB (SUV ± SD)	*p*-Value
Control	0.345 ± 0.025	0.436 ± 0.021	0.471 ± 0.031	0.013 *	0.367 ± 0.027	0.455 ± 0.023	0.514 ± 0.034	0.023 *
				0.006 ′				0.004 ′
				0.375 ^#^				0.181 ^#^
Glioma	0.381 ± 0.021	0.424 ± 0.012	0.436 ± 0.014	0.086 *	0.372 ± 0.028	0.463 ± 0.016	0.440 ± 0.018	0.009 *
				0.047 ′				0.059 ′
				0.511 ^#^				0.338 ^#^
4RT	0.355 ± 0.028	0.455 ± 0.024	0.462 ± 0.015	0.010 *	0.354 ± 0.032	0.475 ± 0.028	0.478 ± 0.017	0.007 *
				0.002 ′				0.002 ′
				0.826 ^#^				0.930 ^#^
AD	0.360 ± 0.039	0.420 ± 0.039	0.435 ± 0.025	0.290 *	0.397 ± 0.038	0.452 ± 0.038	0.460 ± 0.024	0.321 *
				0.118 ′				0.175 ′
				0.740 ^#^				0.848 ^#^

TSPO PET quantification at the group level. Values represent regional group means of the frontoparietal and cerebellar VOIs and their standard deviations. * Specific *p*-value for differences between the tracer uptakes of LABs compared with MABs. ′ LABs compared to HABs. ^#^ MABs compared to HABs. LAB: low-affinity binder; MAB: medium-affinity binder; HAB: high-affinity binder; AD: Alzheimer’s disease; 4RT: 4-repeat tauopathy; SUV: standardized uptake value; and SD: standard deviation.

## Data Availability

The data presented in this study are available upon reasonable request from the corresponding author. The full data are not publicly available due to ethical reasons.

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
