# Peer review of "Impact of TSPO Receptor Polymorphism on [18F]GE-180 Binding in Healthy Brain and Pseudo-Reference Regions of Neurooncological and Neurodegenerative Disorders"

_life, 2021, doi:10.3390/life11060484_

Round 1

Reviewer 1 Report

Overall, this is a well written and structured manuscript. The methodology  could be described in more detail. The discussion is of great importance as it presents a critical look on the demostrated results. I suggest that the authors should deepen the discussion. English editing is also needed.

Reviewer 2 Report

In this study, the influence of the TSPO-genotype on the brain uptake of 18F-GE180 has been investigated. In dependence from the rs6971 genotype, a number of TSPO-targeted PET radioligands can be classified as low, medium, and high affinity binders (LAB, MAB, HAB). So far, none of the available radioligands has been unequivocally proven as ideal and the applicability of TSPO-PET imaging for patients with different binding status remains to be carefully investigated. The current study is a valuable contribution to the assessment of the clinical suitability of 18F-GE180. The susceptibility of 18F-GE180 binding, as measured by standardized uptake value, has been determined in four subject groups – control, glioma, 4RT, AD – with respect to the individual TSPO status – low, medium, high binders. While the uptake in the two pseudo-reference regions in the fronto-parietal and cerebellar hemispheres correlates well with the individuals’ status, the uptake in LABs has been about 20% lower compared to MABs and HABs. Accordingly, the authors suggest not to perform TSPO-PET imaging with 18F-GE180 in patients with neuro-oncological or neurological diseases. In addition to this important result, the herein presented findings indicate the suitability of a reference region based data analyses. The study has been appropriately designed and analysed, the manuscript is very well written, the findings are highly relevant to the field and also the limitations of the study have been announced; however, a few points should be taken into consideration before publication:

- Which hypothesis exactly should be validated/tested by this study?

- line 109 “SUVmax of the tumour”: Mentioning the corresponding PET tracer/protocol perhaps helps readers not fully familiar with the topic.

- In the AD group, contrary to the other subject groups the difference in SUV in LABs is statistically not significant from MABs or HABs. Could the authors please discuss?

- line 256 et seq.: The authors seem to evaluate 18F-GE180 by comparing the extent of differences of binding of various TSPO radioligands towards the different TSPO genotypes in vitro (PBR compounds 17- and 50-fold difference in affinity) with the values obtained herein in vivo (20% different of SUV). However, a proper comparison would require the application of identical experimental methods, such as a head-to-head comparison. Could the authors please discuss?
